# The Order of Grass and Maize Silage Supplementation Modifies Milk Yield, Grazing Behavior and Nitrogen Partitioning of Lactating Dairy Cows

**DOI:** 10.3390/ani9060373

**Published:** 2019-06-19

**Authors:** Ignacio E. Beltrán, Omar Al-Marashdeh, Ana R. Burgos, Pablo Gregorini, Oscar A. Balocchi, Fernando Wittwer, Rubén G. Pulido

**Affiliations:** 1Graduate School, Faculty of Veterinary Sciences, Universidad Austral de Chile, P.O. Box 567, Valdivia, Chile; Ignacio.beltran.gonzalez@gmail.com (I.E.B.); anitarafaela1986@gmail.com (A.R.B.); 2Department of Agricultural Sciences, Lincoln University, P.O. Box 7647, Lincoln, New Zealand; Omar.Al-Marashdeh@lincoln.ac.nz (O.A.-M.); pablo.gregorini@lincoln.ac.nz (P.G.); 3Animal Production Institute, Faculty of Agricultural Sciences, Universidad Austral de Chile, P.O. Box 567, Valdivia, Chile; obalocch@uach.cl; 4Veterinary Clinical Sciences Institute, Faculty of Veterinary Sciences, Universidad Austral de Chile, P.O. Box 567, Valdivia, Chile; fwittwer@uach.cl; 5Animal Science Institute, Faculty of Veterinary Sciences, Universidad Austral de Chile, P.O. Box 567, Valdivia, Chile; 6Instituto de Investigaciones Agropecuarias, INIA Remehue, Ruta 5 Norte, km 8, Osorno 5290000, Chile

**Keywords:** grazing behavior, milk yield, nitrogen excretion, rumen function, supplement allocation

## Abstract

**Simple Summary:**

Herbage growth is reduced during autumn, causing low mass pasture with a high N content and low energy content, while decreasing milk production and increasing urine N (N) excretion. The order of silage supplementation has been suggested as a strategy to improve intake, milk production and reduce urine N excretion, in response to changes in grazing behavior and nutrient intake produced by supplements. This study evaluates the effects of offering grass silage (GS) or maize silage (MS) in the morning or afternoon on milk yield, grazing behavior and N partitioning in lactating dairy cows. We found that time of MS and GS allocation did not modify dry matter intake, however, milk production and urine N excretion was greater for cows receiving MS in the morning and GS in the afternoon compared with cows receiving MS and GS in the morning and afternoon. Results suggest that profitability (high milk production) goes against the environmental goals (low N excretion) under the conditions of the current experiment.

**Abstract:**

The aim of this study was to evaluate the effects of the order of grass silage (GS) and maize silage (MS) supplementation on milk yield, grazing behavior and nitrogen (N) partitioning of lactating dairy cows during autumn. Thirty-six Holstein-Friesian dairy cows were randomly assigned to one of three treatments, and cows remained on these treatments for a 62 days period: (1) MIX; cows supplemented with 3 kg of dry matter (DM) of silage containing 1.5 kg DM of MS and 1.5 kg DM of GS in both the morning and afternoon; (2) GS-MS; cows supplemented with 3 kg DM of GS in the morning and 3 kg DM of MS in the afternoon; (3) MS-GS; cows supplemented with 3 kg DM of MS in the morning and 3 kg DM of GS in the afternoon. All cows received a pasture allowance of 17 kg DM/cow/d and 3 kg DM of concentrate. Grazing time and pasture intake were unaffected by treatment; however, milk production was greater for MS-GS, while milk protein was greater for GS-MS. Urinary N excretion was greater for MS-GS than MIX. In conclusion, MS-GS resulted in high milk yield but also high urinary N excretion, while MIX resulted in low urinary N excretion but also decreased milk yield.

## 1. Introduction

Grazed herbage in temperate regions often supplies nitrogen (N) in excess of animal requirement for milk yield, resulting in low N use efficiency (NUE) and high proportion of dietary N being excreted in the urine, which contributes to the greenhouse gases effects [1]. In addition, seasonal variations in herbage growth rate and nutrient composition occur, causing herbage intake (nutritive quality and quantity) to be insufficient to meet the animal requirements during periods such as autumn-winter [2]. Supplementation has been suggested as a strategy to improve dry matter intake (DMI), milk production and N partitioning. Supplementation with grass silage (GS) has been shown to increase total DMI and milk production [3,4], however, its rapid and extensive N conversion into ammonia (NH_3_) in the rumen [5] suggests a low NUE and high N excretion in grazing dairy systems. Maize silage (MS) poses an alternative to GS to improve DMI and milk production [6]. However, its lower N content [7] and greater content of rumen degradable carbohydrates supplies compared with GS could also enhance rumen N utilization and reduce N excretion. Ruiz-Albarrán, et al. [6] compared both GS and MS, showing a greater DMI, milk production and milk protein in grazing dairy cows supplemented with MS compared to those supplemented with GS, without information about nitrogen partitioning.

The milk production response to supplementation not only depend on composition and amount of supplement offered, but also when the supplement is offered [8], in response to its effect on grazing behavior. According to Adams [9], grazing activity can be modified by offering supplementation at the time of the day where cows spend more time grazing, i.e., early morning and afternoon [10]. Few studies have evaluated the timing of supplementation under restrictive grazing conditions, with positive, negative and non-effects on DMI, milk production and urine N excretion [11,12]. However, just one study has evaluated the timing of supplementation under non-restrictive grazing conditions [13], reporting a greater pasture DMI and milk production when pasture and an energy supplement were offered in the afternoon.

The former studies have evaluated the timing of energy supplements allocation, which modify the grazing behavior due to neuroendocrine factors resulting from the digestion of feed, rather than physical factors, without information about the effects of physical factors of supplementation on grazing behavior. In this way, GS is characterized for a results in greater rumen fill than MS [14], suggesting that GS could affects grazing behavior due to cessation of pasture intake as a consequence of physical factors, as suggested by Sheahan, et al. [15]. In this way, GS allocation before morning grazing could reduce morning grazing time, increasing afternoon grazing, where pasture is characterized for a lower crude protein (CP) and higher water-soluble carbohydrates (WSC) (CITAS). In addition, the best chemical composition of herbage could be improved if maize silage is allocated before afternoon grazing, which is characterized as having a greater WSC and lower CP than grass and grass silage [6,16]. The described above would lead to changes in the nutrient flow through the days, especially the energy and N flow, which could modify rumen fermentation parameters, nitrogen partitioning and milk production.

Therefore, the manipulation of the order of grass silage and maize silage presentation (before morning or afternoon grazing) could be a strategy to modify grazing behavior due to both physical and neuroendocrine factors affecting pasture intake, which also could modify nutrient intake, milk production and nitrogen partitioning. The aim of this study was to evaluate the effects of the order of GS and MS supplementation on milk yield, grazing behavior and N partitioning of lactating dairy cows grazing on an autumn pasture.

## 2. Materials and Methods

This trial was carried out at the Experimental Research Station (ERS) of the Universidad Austral de Chile (latitude 39°47′ S and longitude 73°l4′ W) over a period of 62 days (7th May to 1st July in 2015). The temperature during the period of study averaged 9.3 ± 0.46 °C and daily minimum and maximum temperatures averaged 6.5 °C and 12.8 °C, respectively. Total rainfall was 379 mm. All procedures in this experiment were approved by the Animal. Welfare Committee of Universidad Austral de Chile (69-2012).

### 2.1. Experimental Design

Thirty-six Holstein × Friesian cows, including three rumen cannulated cows, were grouped according to milk production (23.6 ± 4.7 kg), body weight (BW; 509 ± 70 kg) and days in milk (DIM; 60 ± 22). The groups were randomly allocated to one of three treatments: 1) MIX; cows supplemented with 1.5 kg DM of MS and 1.5 kg DM of GS in the morning and again in the afternoon; 2) GS-MS; cows supplemented with 3 kg DM of GS in the morning and 3 kg DM of MS in the afternoon; 3) MS-GS; cows supplemented with 3 kg DM of MS in the morning and 3 kg DM of GS in the afternoon.

Silage supplementation was offered to each cow individually at 09:00 h and 16:00 h (morning and afternoon supplementation, respectively), when cows were held in individual feeding pens for 40 minutes. The GS was harvested from perennial ryegrass swards on October 2014, and then wilted over 24 h before bailing and storing it in plastic bags. Maize silage was rolled, sealed and stored in a pit at the same day of harvesting at the ERS (March 2015). Pasture and grass silages allocation of treatments are summarized in Figure 1.

All cows were offered 3 kg DM of concentrate daily, divided between two equal meals, each of 1.5 kg DM at each milking (08:00 and 15:00 h), using automatic feeders in the milking parlor. Concentrate was comprised of (on a % on DM basis) 49.3 maize, 11.5 soybean meal, 30.0 beet pulp, 4.6 beet molasses and 4.5 mineral mix. The refusal of silages and concentrate were measured individually for each cow at each feeding time during the experiment.

Cows remained on the same treatment throughout the experiment with exception of rumen cannulated cows. The three-rumen cannulated dairy cows were allocated separately to one of three treatments in a Latin Square Design, remaining on each treatment for a 14-day period, with rumen sampling conducted during the final 24 h period of each sampling period.

### 2.2. Grazing Management

The experiment was carried out on 20 ha of perennial ryegrass (*Lolium perenne*) dominant sward, which was subdivided in 8 paddocks. Paddocks were not fertilized during the experiment. All treatments were strip-grazed in the same paddock, separated by an electric fence. Fresh pasture allowance (measured above ground level) of 17 kg DM/cow/day was allocated at 17:00 h, using an electric fence to avoid back grazing. Pre- and post-grazing herbage mass were estimated three times per week using a rising plate meter (RPM, Ashgrove Plate Meter, Hamilton, New Zealand). Each estimation considered 100 compressed sward height measurements by walking through the herbage in a “W” pattern. Using a specific equation for autumn grasslands of southern Chile [17], compressed height data (cm) was transformed into kg DM/ha, using the Equation (1): Y = 120X + 350(1)

R^2^ = 0.74

Where:

Y = herbage mass expressed in kg DM/ha

X = average compressed height

Paddock allocation was conducted according to pre-grazing herbage mass availability with a target pre-grazing of 2800 kg DM/ha and post-grazing of 1400 kg DM/ha. Before the experiment, a grazing wedge was established in order to obtain the required pre-grazing values.

### 2.3. Herbage and Supplement Sampling and Analysis

Pre-grazing herbage samples were collected once a week at 17:00 h, before cows entered the new daily strip, cutting at 4 cm height [2]. Each weekly herbage sample were composed by compiled from samples collected Monday, Wednesday and Friday of each week. Concentrate and silages were sampled during days 19, 34 and 50 of the experiment. All samples of herbage, silage and concentrate were frozen at −20 °C and later freeze-dried and ground through 1-mm sieve (Willey Mill, 158 Arthur H, Thomas, Philadelphia, PA, USA) for chemical analysis. Samples were analyzed for DM, CP, acid detergent fiber (ADF), ash (AOAC, 1996), neutral detergent fiber (NDF) [18] and metabolizable energy (ME) [19,20]. pH and ammonia N (NH_3_-N) in silage samples were derived by Official Methods of Analysis (AOAC) [21].

### 2.4. Dry Matter Intake and Grazing Behavior

Pasture DMI was estimated using an indigestible marker [22] for 12 days during weeks 7 and 8 of the experiment. All cows were given paper capsules containing chromium oxide (6 g/d; 68%wt/wt) after each milking, using an oral dispenser. Over the last 6 days, fecal samples were collected twice per day from the rectum of each cow after the time of dosing. Pasture DMI from chromium oxide excretion was calculated as described by Pulido, et al. [2].

Grazing behavior was determined for each cow on days 17 and 34 of experiment. One trained observer was assigned for each group and grazing, ruminating, and idling behavior were recorded for each cow with a frequency of 10 minutes during daylight (08:30 to 19:00 h) and 15 minutes during nighttime (19:00 to 08:30 h).

### 2.5. Milk Performance

Daily milk yield was measured by an electronic milk meter (MPC580 DeLaval, Tumba, Sweden). Milk yield was analyzed using a bulked sample per week.

Milk was sampled once a week after each milking and analyzed for protein, fat and urea using infrared spectroscopy (Milko-scan, System 4300, Foss Electric, Denmark).

### 2.6. Rumen Parameters

Ruminal concentration of volatile fatty acids (VFA) and ruminal NH_3_ were evaluated using the rumen cannulated cows. Individual rumen samples were collected from three locations in the rumen (cranial, ventral and caudal) at 08:00, 10:00, 13:00, 16:00, 20:00, 00:00 and 03:00 hours during days 25, 39 and 53 of the experiment. Immediately after collection of rumen fluid, the samples (from each ruminal site) were bulked and subdivided into two subsamples. A subsample of 4 mL was diluted with formic acid (4:1 ratio) and rumen concentration of VFA was determined by gas chromatography (Shimadzu GC-2010 plus High-end GC, equipped with GC Capillary Column, SGE, BP21 (FFAP)). A second subsample of 10 ml was acidified with 0.2 mL of 50% trichloroacetic acid solution to estimate rumen NH_3_ concentration using spectrophotometry (Spectronic Genesys 5^®^ spectrophotometer, Milton Roy, Ivyland, PA, USA), as described by Bal, et al. [23].

### 2.7. Nitrogen Partitioning

The N partitioning study was carried out between weeks 7 and 8 of the experiment. The same bulked fecal samples that used to estimate the pasture DMI were used to estimate fecal N content using an autoanalyzer LECO FP528, based on the DUMAS method [21]. Feces samples were frozen immediately after collection and freeze-dried for chemical analysis. Thus, fecal and urinary N excretion (g N/d) were calculated using the equations proposed by Whelan, et al. [24].

### 2.8. Statistical Analysis

Milk production, milk composition, grazing behavior and rumen fermentation parameters were analyzed as repeated measures in time using the MIXED procedure of SAS (PROC MIXED, SAS). The model included the fixed effects of treatment, random effect of cows, day of sampling as repeated measurement and interaction between treatment and time of sampling, with cows as experimental unit.

Dry matter intake and N partitioning were analyzed using the PROC MIXED of SAS. The model included the fixed effects of treatment and the random effect of cows.

Chemical composition of pasture and herbage mass were analyzed using the PROC MIXED of SAS. The model included the fixed effects of treatment and day of sampling, and their interaction.

Assumptions of normality and homogeneity residuals were checked graphically using plots of residuals versus fitted values and normal quantile plots.

Comparison between treatments was carried out using the Tukey test. Results were considered significant at *P* < 0.05 and tendency at *P* < 0.1.

## 3. Results

### 3.1. Pasture and Supplements

The chemical composition of herbage and supplements are presented in Table 1. Herbage chemical composition was similar among treatments (*P* > 0.05) for DM, CP, NDF, ADF and ME. The DM content was 34.7%, 43.1% and 87.3% for grass silage, maize silage and concentrate, respectively. The CP content was 54% lower for maize silage than for grass silage and 60% lower for maize silage than for concentrate.

### 3.2. Dry Matter Intake and Grazing Behavior

Pre- and post-grazing herbage mass, DMI and grazing behavior results are presented in Table 2. Pre- and post-grazing mass were unaffected by treatment, averaging 2793 and 1460 kg DM/ha, respectively. Total and herbage DMI were not different among treatment (*P* > 0.05), averaging 14.4 and 5.4 kg DM/d, respectively.

Daily time cows spent grazing was not affected by treatment, averaging 336 min/d. However, grazing time between morning and afternoon milking (08:00–14:45 h) tended to be greater (*P* = 0.08) for MS-GS than GS-MS and MIX (108, 94 and 83 min, respectively). Grazing time between afternoon and morning milking (15:00–07:45 h) was 14% longer in MIX than for GS-MS (*P* < 0.05) but similar between MIX and MS-GS. Idling time was significantly different among treatments (*P* < 0.05), being 10% longer in GS-MS treatment than MS-GS, but similar between GS-MS and MIX.

### 3.3. Milk Production and Milk Composition

The results of milk production and milk composition are presented in Table 3. Milk production was greater (*P* < 0.05) for MS-GS treatment than GS-MS and MIX (22.4, 20.6 and 269 20.4 kg/d, respectively). Milk protein content (%) was greater (*P* < 0.04) in GS-MS than MS-GS, but similar between GS-MS and MIX. Milk fat content and urea concentration were unaffected by treatment, averaging 4.1% and 4.8 mmol/L, respectively.

### 3.4. Rumen Function and Nitroge Partitioning

Results of rumen fermentation parameters and nitrogen portioning are presented in Table 4. Total rumen VFA, acetate, propionate and butyrate concentration did not differ among treatments, averaging 79.2, 45.9, 16.6 and 11.7 mmol/L, respectively.

There were no significant differences in N intake among treatments, averaging at 386g N/d. N excreted in milk and feces were similar among treatments, averaging 101 and 108 g/d, respectively. Urinary N excretion was 23.5% greater in MS-GS than MIX, but was similar between MS-GS and GS-MS. NUE was not affected by treatment, averaging 26%.

## 4. Discussion

The effect of time of offering energy concentrates [13,25] and maize silage [11,26] have been previously evaluated, showing conflicting effects on milk yield and N partitioning. However, this is the first work evaluating the combined effects of two low cost supplement (grass silage and maize silage) on milk yield, grazing behavior and N partitioning.

### 4.1. Dry Matter Intake and Grazing Behaviour

Time that cows spent grazing between morning milking and afternoon milking, tended to be shorter for cows receiving 1.5 kg or 3.0 kg DM of GS in the morning (MIX and GS-MS, respectively) compared with cows receiving 3.0 kg of MS in the morning (MS-GS). This result suggests that changes in the morning grazing behavior are associated with physical factors coming from supplementation, possibly associated with a GS resulting in a greater ruminal fill than MS [14]. Several studies [27,28] have reported a reduced grazing time (especially first grazing bout length) as ruminal fill increased, supporting our assumption.

The time that cows spent grazing between afternoon milking and morning milking was longer for MIX compared to GS-MS, suggesting that afternoon grazing behavior control could be linked with the greater metabolites released from feed digestion of MS and afternoon pasture, which had a greater WSC content than morning pasture. Numerically higher propionate for GS-MS at 20:00 h than other treatments (+15.8%) could support our assumption, because propionate is a glucose precursor (Wertz-Lutz et al., 2006) which is considered an inhibitor of ghrelin, a powerful hormone stimulating food intake [29]. However, ghrelin concentration was not measured in the current experiment. The propionate concentration suggests that GS-MS could have lower ghrelin concentration than MIX between morning and afternoon milking.

Time of allocation of two supplements (GS and MS) modified grazing time both in the morning and afternoon, however, it was not enough to result in a change in pasture intake of dairy cows grazing under non-restrictive conditions. This lack of effects of time of silages allocation on pasture intake could be associated with the high silage supplementation (42% of the diet) and autumn pasture conditions, which could have masked the effects on pasture DMI by a high substitution rate and low daylight available, respectively [2]. Different studies under restrictive grazing conditions have reported greater pasture DMI when MS [26] or concentrate [25] were allocated in the morning, in response to a long fasting time, which increases the motivation for grazing and thereby, increase herbage intake. Therefore, our results show that effects of time of supplement allocation on pasture intake is associated with the grazing management, i.e., a restricted or unrestricted grazing system.

### 4.2. Milk Production and Composition

Although pasture and total DMI were unaffected, milk production was greater for MS-GS than the other treatments, suggesting that time of silages allocation modified the nutrients intake. According to our results, nutrients intake was not affected by treatments, suggesting that the methodology used to estimate pasture DMI was not accurate enough to detect differences in the estimation of nutrient intake, especially for energy intake. Therefore, greater milk production for MS-GS could be associated with a better nutrient intake, which produced a tendency to greater rumen butyrate concentration in this treatment (*P* = 0.1; Table 4), indicative of greater energy supply for milk production than in other treatments. According to Seymour, et al. [30], milk production had a positive relationship with rumen butyrate concentration, in response to greater energy value than major rumen VFA. Energy is the main factor limiting milk production in grazing dairy system, therefore, all extra energy supply should increase milk production, as reported in this study. Our results disagree with those reported by Trevaskis, et al. [13], where afternoon allocation of pasture and energy-concentrate increased both DM intake and milk production. This lack of concordance is probably associated with the different feeding management processes between studies; Trevaskis, et al. [13] evaluated timing of allocation of one supplement, while our experiment evaluated two supplements allocated at different times.

The lower milk protein in MS-GS than in other treatments could be a dilution effect in response to its greater milk yield [13], because it has been observed that milk protein percentage can increase if milk yield is decreased [31]. However, milk fat was unaffected by treatment, possibly associated with the similar rumen acetate: propionate ratio among treatments [32].

### 4.3. Nitrogen Partitioning and Rumen Function

Treatment did not modify the N intake, in response to similar diet CP content and pasture DMI between treatments. Fecal N excretion was not different between treatments, which may be associated with the high potential degradability of pasture (> 94% for DM and > 96% for CP, unpublished data from this experiment) and GS (>88% for DM and > 90% for CP, unpublished data from this experiment) in the current experiment, which make up more than 58% of the diet across treatments in this study. This resulted in a lower indigestible N intake, which is the most important component of fecal N together with endogenous N [24].

The urinary N excretion was greater in MS-GS than MIX but similar to GS-MS. Numerically greater N intake in MS-GS could explain its greater urinary N excretion, because greater N intake has been associated with greater urinary N excretion, without an increase in fecal or milk N [1,24].

The greater N urine excretion and numerically greater N intake for MS-GS was not reflected in a greater rumen NH_3_ concentration, suggesting that extra N intake for MS-GS was converted into rumen microbial protein, possibly associated with a greater energy supply than other treatments. This would lead to the tendency for greater rumen butyrate observed by MG-GS treatment. However, this increased rumen microbial protein would have been in excess to dietary requirements and therefore excreted as urinary N as reported by Trevaskis, et al. [13].

It was not possible to reduce urinary N excretion and increase milk yield in the same treatment. These results agree with Gregorini, et al. [33], who reported that the number of diet alternatives that allow both the reduction of urinary N excretion and increase or maintain milk yield are small under systems based on grazed pasture. It is possible that the use of MS in combination with GS reduced the potential effects of MS on N and energy intake and thereby masked any effects of MS on N excretion and milk yield. Therefore, more research evaluating the effects of allocation time of supplements with low CP and high energy content on milk yield, grazing behavior and N partitioning is necessary.

## 5. Conclusions

The order of GS and MS allocation allowed for a change in the milk yield, grazing behavior and N partitioning. However, it was not possible to increase milk yield and reduce urinary N excretion in the same treatment. MS-GS resulted in high milk yield but also high urinary N excretion, while MIX resulted in low urinary N excretion but also decreased milk yield. These results suggest that profitability goals (increased milk production) go against the environmental goals (reduced urinary/fecal N) under the conditions of the current experiment.

## Figures and Tables

**Figure 1 animals-09-00373-f001:**
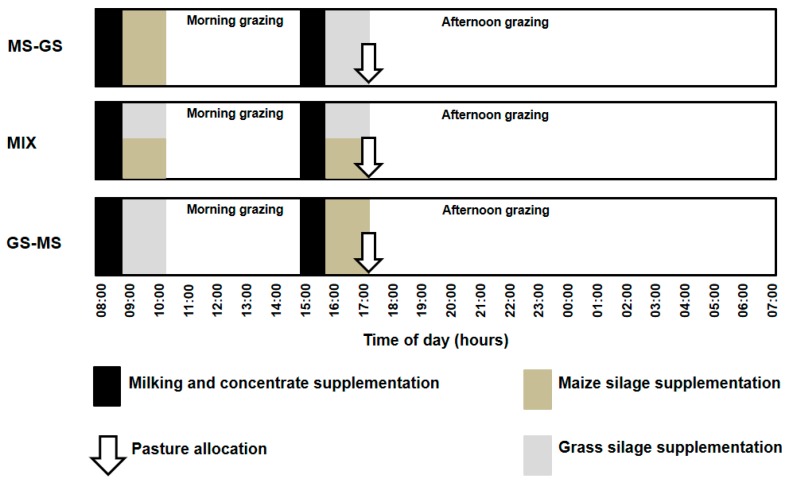
Pasture, grass silage, maize silage and concentrate allocation of treatments. MIX: cows supplemented with 1.5 kg DM of MS and 1.5 kg DM of GS in the morning and again in the afternoon; GS-MS: cows supplemented with 3 kg DM of GS in the morning and 3 kg DM of MS in the afternoon; MS-GS: cows supplemented with 3 kg DM of MS in the morning and 3 kg DM of GS in the afternoon.

**Table 1 animals-09-00373-t001:** Chemical composition of herbage and supplements offered to grazing dairy cows supplemented with 50:50 grass and maize silage in the morning and afternoon (MIX), grass silage in the morning and maize silage in the afternoon (GS-MS) or maize silage in the morning and grass silage in the afternoon (MS-GS).

^1^ Item	Herbage			SEM	Silage				Concentrate	SE
	MIX	GS-MS	MS-GS		Grass	SEM	Maize	SE		
DM	12.9	12.9	13.2	0.2	34.7	4.3	43.1	0.2	87.3	0.05
CP	24.3	25.0	25.5	0.2	13.7	2.3	6.3	0.3	16.0	0.55
NDF	51.6	52.1	51.7	0.4	55.8	1.2	48.5	2.5	20.3	1.20
ADF	24.1	24.8	23.7	0.3	34.4	0.8	25.8	1.3	10.0	0.16
ME	2.7	2.8	2.8	0.01	2.6	0.0	2.7	0.0	3.1	0.03
Ash	10.0	9.9	9.8	0.1	9.7	0.7	3.9	0.1	7.7	0.28
WSC	7.4.	7.02	7.24	0.13	-	-	-	-	-	-
pH	-	-	-	-	4.5	0.1	3.8	0.1	-	-
N-NH_3_	-	-	-	-	10.8	2.6	5.7	0.5	-	-

^1^ DM, dry matter (%); CP, crude protein (% DM); NDF, neutral detergent fibre (% DM); ADF, acid detergent fibre (% DM); ME, metabolizable energy (Mcal/kg DM); WSC, water soluble carbohydrates (% DM); N-NH_3_, ammonia-N (%).

**Table 2 animals-09-00373-t002:** Herbage mass, dry matter intake and grazing behavior and milk production of grazing dairy cows supplemented with 50:50 grass and maize silage in the morning and afternoon (MIX), grass silage in the morning and maize silage in the afternoon (GS-MS) or maize silage in the morning and grass silage in the afternoon (MS-GS).

Item	Treatments			SEM	*P*-Value
	MIX	GS-MS	MS-GS		
Pre-grazing herbage mass (kg DM/ha)	**2803**	2760	2817	41	0.6
Post-grazing herbage mass (kg DM/ha)	1459	1468	1451	22	0.9
Herbage intake (kg DM/ha)	5.2	5.1	5.8	0.26	0.49
Maize silage intake (kg DM/ha)	3.0	3.0	3.0	-	-
Grass silage intake (kg DM/ha)	3.0	3.0	3.0	-	-
Concentrate intake (kg DM/ha)	3.0	3.0	3.0	-	-
Total intake (kg DM/ha)	14.2	14.1	14.8	0.26	0.49
Grazing behavior					
Total grazing time (min/d)	341	315	352	13.3	0.16
Grazing time between 08:00–14:45 h (min)	83	94	108	7.1	0.08
Grazing time between 15:00-07:45 h (min)	257a	221b	244ab	9.3	0.03
Total ruminating time (min)	459	432	446	13.1	0.35
Idling time (min)	551ab	608a	547b	16.6	0.03

Means within a row with different letters differ (*P* < 0.05).

**Table 3 animals-09-00373-t003:** Milk production and milk composition of grazing dairy cows supplemented with 50:50 grass and maize silage in the morning and afternoon (MIX), grass silage in the morning and maize silage in the afternoon (GS-MS) or maize silage in the morning and grass silage in the afternoon (MS-GS).

Item	Treatments			SEM	*P*-Value
	MIX	GS-MS	MS-GS		
Milk production					
Milk yield (kg/d^)^	20.4b	20.6b	22.4a	0.57	0.04
Milk Fat (%)	4.12	4.26	4.06	0.15	0.64
Milk Protein (%)	3.15ab	3.30a	3.06b	0.06	0.04
Milk Urea (mmol/L)	4.97	4.82	4.63	0.31	0.75

Means within a row with different letters differ (*P* < 0.05).

**Table 4 animals-09-00373-t004:** Rumen parameters (VFA, N-NH_3_ and rumen pH) and N partitioning (N intake, milk N, urine N and feces N) of grazing dairy cows supplemented with 50:50 grass and maize silage in the morning and afternoon (MIX), grass silage in the morning and maize silage in the afternoon (GS-MS) or maize silage in the morning and grass silage in the afternoon (MS-GS).

Item	Treatments			SEM	*P*-Value
	MIX	GS-MS	MS-GS		
Rumen parameters					
N-NH_3_ (mmol/L)	10.5	10.4	11.8	0.86	0.38
Acetate (mmol/L)	45.3	44.0	48.3	1.93	0.23
Propionate (mmol/L)	15.9	16.6	17.2	0.88	0.27
Butyrate (mmol/L)	11.6	11.1	12.5	0.39	0.10
Total VFA^2^ (mmol/L)	77.6	76.8	83.3	3.00	0.19
N partitioning					
N intake (g/d)	356	412	389	11.4	0.16
Milk N (g/d)	97	106	100	2.7	0.39
Urinary N excretion (g/d)^1^	150b	196a	185ab	7.4	0.05
Faecal N excretion (g/d)	110	110	103	2.8	0.52
Milk N/N intake (%)	27.3	25.9	25.9	0.5	0.45

^1^ Volatile fatty acid. Means within a row with different letters differ (*P* < 0.05). ^2^ Total VFA = acetate + butyrate + propionate + isobutyrate + valerate + isovalerate.

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
