# Peer review of "The Order of Grass and Maize Silage Supplementation Modifies Milk Yield, Grazing Behavior and Nitrogen Partitioning of Lactating Dairy Cows"

_animals, 2019, doi:10.3390/ani9060373_

Round 1

Reviewer 1 Report

This is a well designed and interesting study.  The manuscript is well written with only a few grammatical issues, a few of which result in some confusion about the context (see comments below).  While the results, discussion and conclusions follow logically the final sentence in the manuscript (lines 278-279) stand out somewhat.  This is mainly because the authors had not previously distinguished between profitability goals (increased milk production) and environmental goals (decreased urinary/fecal N).  See comments below.

Specific comments/suggestions:

Line 72-73: "In this way, GS is characterized for a results in greater rumen fill than MS, suggesting that GS could affects grazing behavior...."

Line 124: "Each Weekly herbage samples were was composed by compiled from three sampling days samples collected Monday, Wednesday, and Friday of each week."

Line 130: "...derived by the method of AOAC AOAC method..."

Line 136: "...described by [2]…" I think you should use the actual citation here, i.e. "...described by Pulido et al. [2]."

Line 167: "...and time of sampling, being the with cows as..."

Line 200-201:  Remove comma after "Although" and remove period after "experiment".

Line 220 - The clause of the sentence starting at line 220; "...suggesting the methodology used was not accuracy to detect differences...."  is confusing.  First it is unclear which method is bein referred to and secondly, "...was not accuracy..." is not proper English grammar.  Some suggestions on this latter point: "...suggesting that the methodology used was not accurate enough...", or ...suggesting that the methodology did not accurately detect…"

Line 258: The sentence, "This supported for the tendency to greater rumen butyrate concentration..." is unclear and not grammatically correct.  Suggest something like this: "This would lead to the tendency for greater rumen butyrate observed in the MS-GS treatment (P=0.1)."

Line 259-261: This sentence is unclear and grammatically flawed.  Suggest it be simplified to something like this:  "However, this increased rumen microbial protein would have been in excess to dietary requirements and therefore excreted as urinary N as reported by Trevaskis et al. [13]. 

Line 278 - 279: Suggest adding parenthetical explanations thus: " These results suggest that profitability goals (increased milk production) go against the environmental goals (reduced urinary/fecal N) under the conditions of the current experiment."

Author Response

Dr. Joe Jacobs

Associate Editor, Animal Production Science

Ref:  animals-501706.

We are very grateful for the opportunity to improve our manuscript “Timing of grass and maize silage supplementation modifies milk yield, grazing behavior and nitrogen partitioning of lactating dairy cows”. All authors accepted Reviewers’ suggestions.

Answers

This is a well-designed and interesting study.  The manuscript is well written with only a few grammatical issues, a few of which result in some confusion about the context (see comments below).  While the results, discussion and conclusions follow logically the final sentence in the manuscript (lines 278-279) stand out somewhat.  This is mainly because the authors had not previously distinguished between profitability goals (increased milk production) and environmental goals (decreased urinary/fecal N).  See comments below.

Specific comments/suggestions:

Line 72-73: "In this way, GS is characterized for a results in greater rumen fill than MS, suggesting that GS could affects grazing behavior...."

·       Response: It was modified as suggested by Reviewer (Line 70).

Line 124: "Each Weekly herbage samples was composed by compiled from three sampling days samples collected Monday, Wednesday, and Friday of each week."

·       Response: It was modified as suggested by Reviewer (Lines 146-146).

Line 130:        "...derived by the method of AOAC AOAC method..."

·       Response: It was modified as suggested by Reviewer (Line 152).

Line 136: "...described by [2]…" I think you should use the actual citation here, i.e. "...described by Pulido et al. [2]."

§  Response: It was modified as suggested (Line 158)

Line 167: "...and time of sampling, being the with cows as..."

·       Response: It was modified as suggested by reviewer (Lines 189-190).

Line 200-201:  Remove comma after "Although" and remove period after "experiment".

·       Response: It was modified as suggested by Reviewer (Line 274).

Line 220 - The clause of the sentence starting at line 220; "...suggesting the methodology used was not accuracy to detect differences...."  is confusing.  First it is unclear which method is being referred to and secondly, "...was not accuracy..." is not proper English grammar.  Some suggestions on this latter point: "...suggesting that the methodology used was not accurate enough...", or ...suggesting that the methodology did not accurately detect…"

·       Response: It was modified as suggested by Reviewer (Lines 290-292).

Line 258: The sentence, "This supported for the tendency to greater rumen butyrate concentration..." is unclear and not grammatically correct.  Suggest something like this: "This would lead to the tendency for greater rumen butyrate observed in the MS-GS treatment (P=0.1)."

·       Response: It was modified as suggested by Reviewer (Line 321).

Line 259-261: This sentence is unclear and grammatically flawed.  Suggest it be simplified to something like this:  "However, this increased rumen microbial protein would have been in excess to dietary requirements and therefore excreted as urinary N as reported by Trevaskis et al. [13].

·       Response: It was modified as suggested by Reviewer (Lines 321-323).

Line 278 - 279: Suggest adding parenthetical explanations thus: " These results suggest that profitability goals (increased milk production) go against the environmental goals (reduced urinary/fecal N) under the conditions of the current experiment."

Response: It was modified as suggested by Reviewer (Lines 336-338).

Sincerely yours 

Rubén G. Pulido

Reviewer 2 Report

The authors manipulate  the order (not timing) of silage presentation and examine this effect on  milk production, grazing, and nitrogen partitioning. The authors  suggest the order of silage type presentation has an effect on grazing,  nitrogen partitioning, and milk production, however; there are serious  statistical shortcomings that prevent clear assessment of these results.  The paper is perhaps interesting to readers in dairy management, but  the impact of the paper is hard to evaluate because the conclusions  drawn are not substantiated by the results, and consequently statistical  analyses presented. There is substantial research in area of silage  supplementation and dairy production as dutifully cited by the authors,  and it is unclear what new knowledge is gained from the present study.

Major concerns:

Importance:  why is it important understand the how supplementation affects cattle  under non-restrictive grazing conditions? Are dairy farms that have  non-restrictive grazing conditions more common or representative of  current farming practices? Is there a policy, economic or special  interest group push to house cattle under non-restrictive grazing  conditions. The authors need to articulate why this incremental study is  important in and adds value to the studies that already assess the  effects of timing supplementation on cattle grazing, milk yield, and  Nitrogen (N) partitioning (refs. 6, 9-13).

·      Similarly,  it is unclear why the authors would expect differences in Nitrogen  partitioning, rumen parameters, and nitrogen partitioning among the  treatment groups- all animals were fed the same diet? The fecal samples  were not examined after each feed type, but were collectively sampled  after the cattle had consumed maize and grass silage. The provisioning  of the same supplement and uniform sample collection method would ensure  that all treatment groups would have the same nitrogen partitioning,  rumen parameters, and milk production. In order to make a distinction  between supplement type and these parameters, the authors would have to  correlate the provisioning of the supplement with fecal collection, and  also ensure the rumen parameters could change under such time courses.

·      The  authors should better describe their study as it relates to the  variables manipulated - the order that the silage was presented, not the  timing of silage presentation. The timing that the silage was presented  was the same across treatment groups (0800 and 1500h), but the order in  which the silage was presented varied across groups. The order of  silage presentation included 1) Grass Silage followed by Maize Silage,  2) Maize Silage followed by Grass silage, or 3) simultaneous  presentation of Maize and Grass. Additionally, in the discussion they  should do a more in depth comparison of their results to previously  published literature. For example, how do the milk production means in  the present study relate to others that manipulated other aspects of  silage presentation?

Analysis & Reporting:

The authors include a lot of results. I have concerns with how the data were analyzed and reported.

·      The  authors did multiple ANOVAs (>20), how did they correct for multiple  comparisons? Are their p-values inflated?  Would corrections for  multiple comparisons eliminate the significant differences in milk yield  (kg/d), milk protein (%), Grazing  time between 15:00-07:45 h (min), and Idling time (min).  The authors  should also report effect sizes (e.g., eta-squared, omega-squared) for  their results to better substantiate the effect of the different  experimental regimes on the measured variables. Prior to applying the  ANOVA did the authors confirm that the residuals did not violate the  assumptions of the ANOVA?

·      Each mean should have a SEM; what is the significance of a composite SEM written in the Tables?

·      Why  do the authors present some data in the discussion but not fully in the  results (lines 243-250)? Presenting the results for N in the discussion  with out disclosing the data for full peer-review evaluation is  unacceptable. If the authors want to discuss findings the authors should  include the results. If the authors choose not to present the results  and statistical analysis for findings, then these findings should be  removed.

·      The  authors should parse out the milk analysis by time of collection, so  more detailed relationships between silage order provisioning and milk  production can be assessed.

Integration of hypotheses results and discussion:

·      Better  integration of hypotheses, results and discussions. The authors suggest  that GS will lead to greater rumen fill, and thus decrease grazing  time, however. The authors found that the mix had the lowest grazing  time in the morning, and the MS cows fed in the afternoon had less  grazing activity. 

·      Why  did the authors not mention the results for GS-MS in abstract? The  omission of the results for one of their treatment groups is a glaring  gap in their abstract.

Figures:

·      To  enhance the understanding of the timeline for the data collection, and  treatment groups a figure that graphical presents the experimental  design and sampling regime would enhance the paper.

Minor concerns:

The  authors could enhance the clarity of the manuscript by writing out  abbreviations and initialisms and abbreviations when they are first used  in the text, and alter the wording of some sentences to increase reader  understanding.

What does this mean?

·      Lines 27-29: Rephrase this statement. It is hard to understand.

·      Line  34: What is DM? How is DM different from DMI? What is N? It is  important to spell out abbreviations and initialisms when they are first  used, so that the reader is not confused.

·      Line 134-35: How is the dry matter collected twice per day at time of dosing?

Author Response

Dr. Joe Jacobs

Associate Editor, Animal Production Science

Ref:  animals-501706.

We are very grateful for the opportunity to improve our manuscript “Timing of grass and maize silage supplementation modifies milk yield, grazing behavior and nitrogen partitioning of lactating dairy cows”. All authors accepted Reviewers’ suggestions.

Answers

1) The authors manipulate the order (not timing) of silage presentation and examine this effect on milk production, grazing, and nitrogen partitioning. 2) The authors suggest the order of silage type presentation has an effect on grazing, nitrogen partitioning, and milk production, however; there are serious statistical shortcomings that prevent clear assessment of these results.  The paper is perhaps interesting to readers in dairy management, but the impact of the paper is hard to evaluate because the conclusions drawn are not substantiated by the results, and consequently statistical analyses presented. 3) There is substantial research in area of silage supplementation and dairy production as dutifully cited by the authors, and it is unclear what new knowledge is gained from the present study.

·       Response 1: We agree that we manipulated the order of silage presentation and not timing, therefore, title and objectives were modified as suggested. The new title is “The order of grass and maize silage supplementation modifies milk yield, grazing behavior and nitrogen partitioning of lactating dairy cows”.

Response 2: We agree that there is information about time of concentrate and maize silage supplementation. However, all studies found in the literature have focused on the  manipulation of  the order of just one energy supplement (concentrate or maize silage), which only could modify the grazing behavior in response to chemical factors associated with the supplement.

Response 3: We agree that there is substantial research in area of silage supplementation and dairy production; however, ours results suggest that changes in the chemical and physical factors that modify grazing behavior also have an important effect on nutrient intake, rumen function, milk production and nitrogen partitioning.  These also allow suggesting a new strategy to manipulate the order of supplementation of two silage during the day.

Major concerns:

Importance:  1) why is it important to understand how supplementation affects cattle  under non-restrictive grazing conditions? 2) Are dairy farms that have non-restrictive grazing conditions more common or representative of current farming practices? Is there a policy, economic or special  interest group push to house cattle under non-restrictive grazing  conditions. 3) The authors need to articulate why this incremental study is important in and adds value to the studies that already assess the effects of timing supplementation on cattle grazing, milk yield, and Nitrogen (N) partitioning (refs. 6, 9-13).

·       Response 1:  In this study, a restricted herbage allowance (17 kg DM/cow/day) was used (measured at ground level). It is accepted that non-restrictive values of herbage allowance would be over 30 kg MS/cow/day. This grazing situation with a restrictive supply of pasture is most frequent in autumn; when the requirements of the dairy herd exceeds the rate of prairie growth due to climate conditions. In addition, in this dairy production system is common that cows spend most of the time grazing between milking times; also the daylight hours are shorter than in spring and summer. Therefore, it is important to know how supplementation affects food intake, animal response and metabolism in dairy cows grazing this type of pastures.

·       Response 2: The condition of non-restricted grazing occurs mainly during the spring-summer season, a period in which the rate of growth of the pasture exceeds the consumption requirements of the dairy herd.  During autumn season, this condition of non-restricted grazing is very rare. Under non-restrictive grazing condition that occur in spring-summer, there is no interest in the productive system to confine the cows and in general, they are managed at grazing. During the period of lower pasture growth, autumn-winter, when grazing is restricted, the partial confinement of the animals with supplementary feed, especially silages, is more frequent.

·       Response 3: The previous studies provided great knowledge to the effect of several  parameters related to milk production, metabolism and grazing behavior of cattle during autumn. However, the aim of this study was to evaluate the effects of the order of GS and MS supplementation on milk yield, grazing behavior and N partitioning of lactating dairy cows, when graze a new strip of pasture in the afternoon under restrictive grazing conditions.

Similarly,  it is unclear why the authors would expect differences in Nitrogen  partitioning, rumen parameters, and nitrogen partitioning among the  treatment groups- all animals were fed the same diet? The fecal samples  were not examined after each feed type, but were collectively sampled  after the cattle had consumed maize and grass silage. The provisioning of the same supplement and uniform sample collection method would ensure that all treatment groups would have the same nitrogen partitioning,  rumen parameters, and milk production. In order to make a distinction between supplement type and these parameters, the authors would have to  correlate the provisioning of the supplement with fecal collection, and  also ensure the rumen parameters could change under such time courses.

·       Response: We suggest that changes on grazing, nitrogen partitioning, and milk production, in response to manipulation of the order of two silages could modify grazing behavior. Specifically, changes in the time of the day when cows are grazing. If the afternoon grazing time is increased, we could increase the pasture intake at the time of day when pasture usually has a greater nutritive value (lower crude protein and higher water-soluble carbohydrates than morning pasture) and thereby, we could modify the nutrient flow throughout the day. Regarding rumen parameters, rumen liquid samples were collected seven times per day of sampling, to be able to study the changes in these parameters because of treatments and thereby, supporting other results such as nitrogen partitioning, milk production and grazing behavior.

The  authors should better describe their study as it relates to the  variables manipulated - the order that the silage was presented, not the  timing of silage presentation. The timing that the silage was presented  was the same across treatment groups (0800 and 1500h), but the order in  which the silage was presented varied across groups. The order of  silage presentation included 1) Grass Silage followed by Maize Silage,  2) Maize Silage followed by Grass silage, or 3) simultaneous  presentation of Maize and Grass. Additionally, in the discussion they should do a more in depth comparison of their results to previously  published literature. For example, how do the milk production means in  the present study relate to others that manipulated other aspects of  silage presentation?

·       Response: We are agreed that we manipulated the order of silage presentation, therefore, it was modified as suggested.

Analysis & Reporting:

The authors include a lot of results. I have concerns with how the data were analyzed and reported.

1) The  authors did multiple ANOVAs (>20), how did they correct for multiple  comparisons? Are their p-values inflated?  Would corrections for  multiple comparisons eliminate the significant differences in milk yield  (kg/d), milk protein (%), Grazing  time between 15:00-07:45 h (min), and Idling time (min).  2) The authors should also report effect sizes (e.g., eta-squared, omega-squared) for  their results to better substantiate the effect of the different  experimental regimes on the measured variables. 3) Prior to applying the  ANOVA did the authors confirm that the residuals did not violate the  assumptions of the ANOVA?

1)    We agree with Reviewer that we ran several models for target variables. We decided to use one model for each target variable based on several published papers, for example, Pulido et al.  (2015, Livestock Science), Al-Marashdeh et al. 2016 (Journal of dairy Sciences), Morales et al. (2018). However, we agree with reviewer that another way to analyze the data is using one model that include all variables.

2)    We agree with the reviewer that it is possible to include the size effect using eta-squared or omega-squared. However, we used p-value because all the literature that refers the evaluation of time of supplement allocation only reported p- value instead of eta-square when a mixed model was used.

3)    We confirmed that the residuals did not violate the assumption (normality and homogeneity of variance) for each variable. We could provide this information if it is necessary.

Each mean should have a SEM; what is the significance of a composite SEM written in the Tables?

·       Response: We used Standard Error of the Mean (SEM) instead of Standard Deviation because We wanted to show the variation around the general mean;  similar as reported in studies that evaluated the time of pasture (Pulido et al., 2015) and supplement allocation (Al-Marashdeh et al., 2016, Mattiauda et al., 2019).

Why do the authors present some data in the discussion but not fully in the  results (lines 243-250)? Presenting the results for N in the discussion with out disclosing the data for full peer-review evaluation is  unacceptable. If the authors want to discuss findings the authors should  include the results. If the authors choose not to present the results and statistical analysis for findings, then these findings should be  removed.

4)    Response: The suggestion was accepted and all results  was included in  the Results Section of the  manuscript   (see Lines 199-251)

The  authors should parse out the milk analysis by time of collection, so  more detailed relationships between silage order provisioning and milk  production can be assessed.

·       Response: We consider that the analyses of milk by time of collection could offer a valuable information; however, in this research, we analyzed milk composition by day, putting average samples of morning and afternoon together, because we were interested in finding the differences in the total daily milk production instead of between milking time. Therefore, it was not considered as objective of this research.

Integration of hypotheses results and discussion:

Better integration of hypotheses, results and discussions. The authors suggest that GS will lead to greater rumen fill, and thus decrease grazing  time, however. The authors found that the mix had the lowest grazing  time in the morning, and the MS cows fed in the afternoon had less  grazing activity.

·       Response: We consider that after all the amended realized to the manuscript; we accomplish a better integration of all the manuscript sections, as the reviewer requested it.

Why  did the authors not mention the results for GS-MS in abstract? The  omission of the results for one of their treatment groups is a glaring  gap in their abstract.

·       Response: it was added in the abstract as suggested (Line 38-39)

Figures:

To enhance the understanding of the timeline for the data collection, and  treatment groups a figure that graphical presents the experimental  design and sampling regime would enhance the paper.

·       Response: It was added as suggested (Figure 1, lines 118-124).

Minor concerns:

The  authors could enhance the clarity of the manuscript by writing out  abbreviations and initialisms and abbreviations when they are first used  in the text, and alter the wording of some sentences to increase reader  understanding.

·       Response: Text was modified as suggested

 What does this mean?

Lines 27-29: Rephrase this statement. It is hard to understand.

·       Response: It was modified as suggested

Line  34: What is DM? How is DM different from DMI? What is N? It is  important to spell out abbreviations and initialisms when they are first  used, so that the reader is not confused.

·       Response: It was modified in the text as suggested (Line 27-28).

Line 134-35: How is the dry matter collected twice per day at time of dosing?

·       Response: Dry matter was collected once finished the time of dosing. Text was modified as suggested (Line 157).

Sincerely yours 

Ruben G. Pulido